# Thermal Evolutions to Glass-Ceramics Bearing Calcium Tungstate Crystals in Borate Glasses Doped with Photoluminescent Eu^3+^ Ions

**DOI:** 10.3390/ma14040952

**Published:** 2021-02-18

**Authors:** Takahito Otsuka, Martin Brehl, Maria Rita Cicconi, Dominique de Ligny, Tomokatsu Hayakawa

**Affiliations:** 1Field of Advanced Ceramics, Department of Life Science and Applied Chemistry, Graduate School of Engineering, Nagoya Institute of Technology, Gokiso, Showa, Nagoya, Aichi 466-8555, Japan; t.otsuka.098@stn.nitech.ac.jp; 2Institute of Glass and Ceramics, Department of Materials Science and Engineering, University of Erlangen-Nuremberg, Martensstraße 5, DE-91058 Erlangen, Germany; martin.brehl@fau.de (M.B.); maria.rita.cicconi@fau.de (M.R.C.); dominique.de.ligny@fau.de (D.d.L.); 3Frontier Research Institute for Materials Science (FRIMS), Nagoya Institute of Technology, Gokiso, Showa, Nagoya, Aichi 466-8555, Japan

**Keywords:** glass-ceramics, calcium tungstates, borate glasses, Eu^3+^ luminescence, asymmetry ratio

## Abstract

Thermal evolutions of calcium-tungstate-borate glasses were investigated for the development of luminescent glass-ceramics by using Eu^3+^ dopant in a borate glass matrix with calcium tungstate, which was expected to have a combined character of glass and ceramics. This study revealed that single-phase precipitation of CaWO_4_ crystals in borate glass matrix was possible by heat-treatment at a temperature higher than glass transition temperature *T*_g_ for (100−*x*) (33CaO-67B_2_O_3_)−*x*Ca_3_WO_6_ (*x* = 8−15 mol%). Additionally, the crystallization of CaWO_4_ was found by Raman spectroscopy due to the formation of W=O double bondings of WO_4_ tetrahedra in the pristine glass despite starting with the higher calcium content of Ca_3_WO_6_. Eu^3+^ ions were excluded from the CaWO_4_ crystals and positioned in the borate glass phase as a stable site for them, which provided local environments in higher symmetry around Eu^3+^ ions.

## 1. Introduction

Monolithic materials of glass-ceramics have recently attracted a lot of interest because of their potential applications as toughened ceramics for biomedical uses [1,2,3], ionic conductors for energy conversions [4,5,6], and luminescence phosphors for efficient illuminations and display devices [7,8,9]. To design the synthesis of glass-ceramics with a targeted crystal, a choice of starting glass composition is to be carefully considered. In this study, the material evolution of glass-ceramic bearing calcium tungstate crystals in a glass matrix doped with red-luminescent Eu^3+^ ions was investigated. We aimed for precipitation of calcium tungstate crystals as a single phase in glasses. Additionally, control in local structures around Eu^3+^ ions simultaneously doped with the initial host glasses was attempted. There are two types of calcium tungstate crystals of interest as rare-earth phosphors [10,11,12], tetragonal CaWO_4_ and monoclinic Ca_3_WO_6_. The former is composed of WO_4_ tetrahedra [13], while the latter has a double perovskite structure with CaO_6_ and WO_6_ octahedra [14]. The question is whether Eu^3+^ ions are finally positioned after the precipitation of crystals in glass-ceramics based on calcium tungstates.

The choice of the host matrix is very important for this material elaboration. Among preliminary screening tests with silicate, borate, phosphate, etc., calcium borate was selected as a host matrix for its compatibility with the calcium tungstate crystals and low melting point. Ca_3_WO_6_ was used as a starting crystal, which is known to exhibit quite high red luminescent purity for Eu^3+^ ions [14]. In this study, glass samples were first prepared from 33CaO-67B_2_O_3_(in molar) glass and Ca_3_WO_6_ crystal with various compositional ratios, and their thermal and structural properties were investigated. Additionally, they were then examined to know which crystal phases were obtained by thermal treatment at a temperature higher than the glass transition temperature. This material elaboration could be utilized to develop luminescent glass-ceramics. Hence, thermal properties of glass transition temperature (*T*_g_) and on-set crystallization temperature (*T*_x_), and melting point (*T*_m_) were here reported, as well as the results of elementary analysis. Structures of the pristine glasses were examined by Raman spectroscopy to elucidate the role of the unit structures of borate and tungstate in glass on the precipitation of calcium tungstate crystals. X-ray diffractometry showed that CaWO_4_ crystals were precipitated in the borate glassy matrix rather than Ca_3_WO_6_, despite higher calcium content in the pristine glasses. Photoluminescence properties of Eu^3+^-doped (100 − *x*)(33CaO-67B_2_O_3_) − *x*Ca_3_WO_6_ glass and glass-ceramics were also surveyed and the spectral change due to the precipitation of calcium tungstate was discussed in light of the asymmetry ratio of Eu^3+^ ions [15] derived from the luminescence intensity of electric dipole ^5^D_0_-^7^F_2_ transition against that of magnetic dipole ^5^D_0_-^7^F_1_ transition, which revealed the thermal evolution of local structures around Eu^3+^ ions to more stable and symmetric ones that were different from CaWO_4_ crystal.

## 2. Experimentals

### 2.1. Glass Synthesis & Crystallization

H_3_BO_3_, CaCO_3_, Eu_2_O_3_, and WO_3_ were used as received for the sample preparation. In this study, (100 − *x*)(33CaO-67B_2_O_3_) − *x*Ca_3_WO_6_ (*x* = 0, 1, 2, 4, 8, 12, 15 and 16 mol%) glasses and 85 (33CaO-67B_2_O_3_) − 15Ca_2.98_Eu_0.02_WO_6_ glasses were synthesized by a melt-quenching method from homogeneous mixtures of Ca_3_WO_6_ or Ca_2.98_Eu_0.02_WO_6_ powder in crystal and 33CaO-67B_2_O_3_ glass powder (denoted as 33CaB). The mixed powder was melted in a platinum crucible by heating at 1100 °C in a furnace for *x* = 1–8 mol% Ca_3_WO_6_ and quenched on a metallic plate. To gain glass samples over *x* = 12 mol%, a higher melting temperature of 1400 °C and water quenching were needed to apply a faster cooling rate.

The details of the respective powders used to prepare the glass samples studied (the pristine) are shown below: 33 CaB glass was synthesized by a melt-quenching method. Firstly, H_3_BO_3_ was heated at 100 °C for 24 h for the evaporation of adsorbed H_2_O molecules. The mixture of baches powder of H_3_BO_3_ and CaCO_3_ was melted in a platinum crucible at 1080 °C for 2 h in a furnace and quenched on a metallic plate. Ca_3_WO_6_ and Ca_2.98_Eu_0.02_WO_6_ powders were synthesized by the traditional solid-state reaction method. A batch powder with a 3.2/1 molar ratio of CaCO_3_/WO_3_ was mixed for 12 h with ethanol and a ZrO_2_ milling ball, and calcined at 1200 °C for 12 h in an alumina crucible, after drying at 80°C [16]. CaWO_6_ powder also gained from the same powder calcination procedure. Ca_0.98_Eu_0.02_WO_4_ powder synthesized by CaCO_3_, WO_3,_ and Eu_2_O_3_ using a solid-state reaction method. The stoichiometric powders were mixed for 30 min with ethanol in a mortar and calcinated at 1100 °C for 6 h in an alumina crucible.

Glass-ceramics with 2–15 mol% Ca_3_WO_6_ contents were gained by a heat-treatment at a temperature higher than their respective glass transition point, 660 °C, 665 °C, and 635 °C applied for 2 and 12, 4 and 8, and 15 mol% samples for 90 h in a furnace, as shown in Table 1.

### 2.2. Characterizations

X-Ray Diffraction (XRD) patterns were obtained with Bruker D8 Advance eco by using Cu Kα radiation to confirm glass formation and for the assignment of crystalline phases that were precipitated. Scanning Electron Microscopy (SEM) and Energy Dispersive X-ray Spectroscopy (EDS) (Quanta 200, FEI Co., Hillsboro, OR, USA) were adopted for sample observations and quantitative elemental analysis. Thermal properties were estimated from Differential Scanning Calorimeter (DSC) (NETZSCH DSC404F1). Raman spectroscopy using a green laser (532 nm) as a light source (Thermo Scientific NicoletTM Almega) was adopted for understanding the chemical unit structures in each of the samples. To investigate the luminescence properties, photoluminescence (PL) spectra were examined by a fluorescence spectrophotometer equipped with double monochromators (Czerny-Turner) in excitation and emission (Fluorolog3, Horiba Jobin Yvon), using a 450 W Xe-lamp as an excitation source.

## 3. Results and Discussion

### 3.1. Quenched Glasses of (100 − x)33CaB-xCa_3_WO_6_ (x = 0~16 mol%)

Figure 1 shows a photo-image of the glass samples and glass-ceramics at *x* = 4 and 8 mol% Ca_3_WO_6_. The clear and transparent glasses were gained in the range between *x* = 0 to 4 mol% Ca_3_WO_6_. The 8 mol% sample appeared partially crystallized, but was at most clear and transparent, as seen in Figure 1. While a normal quenching rate was applied for *x* =0−8 mol% to obtain the transparent glass samples, quit rapid quenching into water (called “water quenching” here) was needed over *x* = 12 mol%. However, even after the water quenching, the obtained glasses at *x* = 12–16 mol% still had a part of crystals. It was because the quenching was conducted so that the crucible with glass melt was put into water from the bottom of the crucible and roughly 80% of the crucible was sunk in the water bath. The top of the melt was not covered with water to avoid unexpected influences of water molecules on the structure of the obtained borate glasses. If the crucible was not completely cooled down to room temperature and then the crucible was taken from the water bath, the supercooled liquid was resultantly kept warm to allow it to be crystallized. Nevertheless, the crystallized portion was only just seen between the interface of glass melt and crucible for *x* = 12 and 15 mol% and it was found to be small. The volume ratio of glass and crystals was about 9:1 for *x* = 12 and 15 mol%, and about 6:4 for *x* = 16 mol%, as shown in Table 1. (Other possibilities will be discussed in Section 3.3. EDS elementary analysis for 33CaB-Ca_3_WO_6_ glasses.) Figure 2 shows an SEM image of the crystal part of the 15 mol% sample of 85(33CaO-67B_2_O_3_) – 15Ca_3_WO_6_. The observed morphology of the crystal is so unique, and the SEM image clarifies the line structure of the surface of the crystal, indicating uniaxial nuclear growth of the tetragonal CaWO_4_ phase, which will be discussed later. Toward further experiments, the crystallized parts for 12 and 15 mol% were carefully removed, and then clear and transparent glassy parts were extracted. However, at *x* = 16 mol%, serious crystallization occurred and thus it was not used for the heat-treatment.

### 3.2. XRD Patterns of 33CaB-Ca_3_WO_6_ Glasses & Glass-Ceramics

Figure 3a shows XRD patterns of the samples (*x* = 0–15 mol%) mentioned above. All XRD patterns shown here have a broad halo characteristic that exhibits their amorphous natures. At *x* = 0 to 15 mol%, the samples are confirmed to be homogeneous without any crystalline components, which were thus used as pristine glass samples for the further heat-treatment. The above glass formation region is in good agreement with one reported in the previous research [17].

After the heat-treatment, the 2 mol% samples were completely devitrified from the surface to the center of the glass, while the 4 and 8 mol% samples heated became opaque just on the surface and white dots appeared on a part of the surface. The 12 and 15 mol% samples became white and appeared to be completely crystallized after the heat-treatment. The impacts of the heat-treatment were firstly examined by XRD methodology.

Figure 3b shows XRD patterns of the samples after the heat-treatment at a temperature higher than *T*_g_ for 90 h (See Table 1). In this figure, a theoretical XRD pattern of the crystal structure of CaWO_4_ (ICSD ID No.15586) calculated by using RIETAN-FP [18] is also shown for comparison. From the XRD patterns, a crystalline phase of CaWO_4_ was found at *x* = 12 and 15 mol% of Ca_3_WO_6_ (the photo-image of them are depicted in Appendix A as Appendix A), as well as at *x* = 15 mol% of Ca_2.98_Eu_0.02_WO_6_. It is noted at *x* = 15 mol% for non- and Eu^3+^ doped samples, the peaks observed at 2θ ~ 31.5° and 65.6°, assigned to 004 and 008 reflections of CaWO_4_ were enhanced in intensity when the patterns were compared with the powder pattern calculated. This result indicates a crystalline orientation to the c-axis for the precipitated CaWO_4_ phase. It can also be mentioned that no crystalline secondary or impurity phases were observed at *x* =12 and 15 mol%. However, many X-ray reflections were detected for the 2 mol% sample after the heat-treatment, which were identified as CaB_4_O_7_ and CaB_2_O_4_ phases, as well as the CaWO_4_ phase. *x* = 4 mol%, XRD data showed an amorphous state of the sample, but a small amount of CaWO_4_ phase was found by Raman spectroscopy, which was shown in Appendix A as Appendix A. Moreover, it should be borne in mind that the single-phase precipitation of CaWO_4_ crystal in borate glasses requires a Ca_3_WO_6_ content higher than 8 mol% against 33CaB glass.

### 3.3. EDS Elementary Analysis for 33CaB-Ca_3_WO_6_ Glasses

Figure 4 shows the glass compositions (B_2_O_3_, CaO, and WO_3_) of the 33CaB-Ca_3_WO_6_ glasses (glass parts) for *x* = 8 to 16 mol%, estimated by EDS measurements. Qualitatively the decrease in B_2_O_3_ content and the increase in CaO and WO_3_ with the addition of Ca_3_WO_6_ are matched with the variations of the theoretical contents. However, the experimental values of CaO and WO_3_ contents became lower than the theoretical ones for the respective components. However, the experimental value of B_2_O_3_ content was higher than the theoretical one. It may be caused by the possible evaporation of CaO and WO_3_ components during melting because the addition of Ca_3_WO_6_ to 33CaB glass required a higher melting temperature of 1400 °C. During the melt preparation, the 33CaB component was relatively faster changed to a liquid phase in the initial stage of melting, where Ca_3_WO_6_ crystals could stay in a solid-state. It can be noted that the deviation between the experimental and theoretical contents was reduced for the higher Ca_3_WO_6_ concentration. A mechanism causing this behavior is still not known, but it could be speculated that the CaO component from Ca_3_WO_6_ crystals was preferentially introduced into the 33CaB melt and then made it easy to further incorporate the remaining WO_3_ component to the melt for the higher Ca_3_WO_6_ concentration, especially *x* = 16 mol%. In Section 3.1, we explained the partial crystallization of the quenched glasses for *x* = 12 to 16 mol% after the water quenching. The fact that the crystals were found, especially between the glass and crucible, it may result from the decomposition of Ca_3_WO_6_ to 2CaO and CaWO_4_. If it is the case, the observation in Section 3.1 could be explained, such that the CaO component was incorporated into 33CaB melt, while the CaWO_4_ was partially melted, but still, CaWO_4_ crystals remained after the water quenching. As an alternative possibility, it is deduced that the CaO and CaWO_4_ produced from the decomposition of Ca_3_WO_6_ crystals could be melt, but the high viscosity of the melt brought about the heterogeneity of the melt, and eventually, a CaWO_4_ rich liquid was free to be precipitated at the lower part of the quenched glass.

### 3.4. Thermal Properties

Figure 5a shows the DSC curves of (100 − *x*) (33CaO-67B_2_O_3_) – *x* Ca_3_WO_6_ glasses. These curves exhibit the typical thermal behavior of glasses with a small endothermic peak, large exothermic peaks, and a large endothermic peak, representing glass transition temperature (*T*_g_), on-set crystallization temperature (*T*_x_), and melting temperature (*T*_m_), respectively. The values of *T*_g_ and *T*_x_ as a function of Ca_3_WO_6_ concentration are shown in Figure 5b. The value of *T*_g_ slightly decreased from ~640 °C to ~630 °C, up to 12 mol% Ca_3_WO_6_ concentration and then dropped down to ~600 °C at 15 mol% Ca_3_WO_6_. The value of *T*_x_ increased from 760 °C to ~785 °C between 0 and 2 mol% Ca_3_WO_6_. For higher Ca_3_WO_6_ concentrations, the *T*_x_ value decreased drastically to ~740 °C until 4 mol% Ca_3_WO_6_ and then decreased more gently to ~700 °C for the further higher Ca_3_WO_6_ concentrations.

The parameter T (=*T*_x_ − *T*_g_) and Hruby parameter, *K*_gl_ given by Reference [19]
(1)$$K_{gl}=\frac{T_{x}-T_{g}}{T_{m}-T_{x}}$$

This is shown in Figure 5c to evaluate their glass stability. ΔT represents the temperature interval during nucleation. Thus, ΔT is also used to evaluate the stability of the glassy state against crystallization. The values of ΔT at 1–2 mol% Ca_3_WO_6_ concentration are higher than 100 °C, indicating that they are more stable against devitrification [20]. However, the ΔT values at Ca_3_WO_6_ concentrations over 12 mol% were lower than 80 °C, indicating that these Ca_3_WO_6_-rich glasses are easily crystallized in comparison to the cases of the lower Ca_3_WO_6_ concentrations. That is well-matched with the fact that CaWO_4_ single-phase appeared by crystallization at 12 and 15 mol% Ca_3_WO_6_. *K*_gl_ also showed a similar tendency. The glass-forming ability is in general determined by measuring the critical cooling rates. Firstly, to be mentioned, it is proved that, for thermally stable glass-forming systems, the value of *K*_gl_ is more than 0.1 and hence thermal stabilization of glasses can be characterized by high values of *K*_gl_ and vice versa [21]. From previous experimental and theoretical researches on the various glass-forming compositions where the critical cooling rate was correlated with the glass stability (*K*_gl_), it is also found that there exists an empirical relation between *K*_gl_ and critical cooling rate [22,23]. Therefore, it is plausible to use *K*_gl_ for judging the glass-forming ability [24]. The Ca_3_WO_6_ concentration dependence of *K*_gl_ is shown in Figure 5c. The values, *K*_gl_ of 12, 15 mol% Ca_3_WO_6_ are lower than *K*_gl_ of 1–8 mol% Ca_3_WO_6_, indicating that 12, 15 mol% Ca_3_WO_6_ have the low glass-forming ability. This is corresponding to the fact that the 12, 15 mol% samples needed a higher cooling rate, as using water to quench the glass melt, than the 1–8mol% samples. The higher and lower values of ΔT and *K*_gl_ at the 1, 2 mol% and 12, 15 mol% Ca_3_WO_6_ seem influenced by the same factor: The introduction of WO_3_ as a network former to B_2_O_3_ glass network at the lower concentration would reinforce the resultant glass structures, however, the influence of CaO as an inhibitor to glass-forming must become dominant over WO_3_ as a network former at the samples with 8–16 mol% Ca_3_WO_6_.

### 3.5. Raman Spectroscopy

Figure 6a,b show area normalized Raman spectra of 0–16 mol% Ca_3_WO_6_ glass samples and deconvoluted Raman spectra of 33CaO-67B_2_O_3_ glass. The peaks at 420–485 cm^−1^ (Nos.2 and 3), 630 cm^−1^ (No.4), 755–770 cm^−1^ (No.6), and 950 cm^−1^ (No.8) can be assigned to B–O stretching modes of borate symmetric and asymmetric vibrations, ring-type metaborate, four- and three-coordinated boron in diborate [25] and orthoborate group, respectively. Additionally, the peaks at 300–350 cm^−1^ (No.1) and 850–920 cm^−1^ (No.7) are assigned to the O-W-O bending mode in distorted γ–WO_6_ and W–O stretching vibrations of W^6+^ = O double bonding [26], respectively. The deconvoluted peak intensity of Nos.1, 7, and 8, and Nos.2, 3, 4, and 6 as a function of Ca_3_WO_6_ concentration showed in Figure 7a,b, respectively. The peak intensities of Nos. 2, 3, 4, and 6 decrease with Ca_3_WO_6_ concentration, whereas the peak intensity of No.8 increases with the Ca_3_WO_6_ concentration. The decreasing peak intensities of Nos.2 and 3 indicate the decrease amount of boron in the glass host. This result corresponds to the decreasing number of B_2_O_3_ estimated by EDS. The borate ring groups, ring-type metaborate, and diborate have bridging oxygen and work as a glass network former. Orthoborate groups have non-bridging oxygens and are located at the end of glass networks. The above result that bridging oxygen and non-bridging oxygen decreased and increased with Ca_3_WO_6_ concentration, respectively, shows the host glass structures were strongly affected by Ca^2+^ ions working as a network modifier with the Ca_3_WO_6_ additions. Thus, the glass sample with lower Ca_3_WO_6_ concentration included much of the amount of borate ring structures in comparison to the samples with higher Ca_3_WO_6_ concentration, and it seems that an impurity phase CaB_4_O_7_, which includes a borate ring structure, appeared in the crystallized 2 mol% Ca_3_WO_6_ sample.

However, the peak intensity of Nos.1 and 7 exhibited an increasing tendency with Ca_3_WO_6_ concentration. Especially, peak No.7 showed a significant increase in comparison with peak No.1. In other words, a more significant change in the peak intensity about W = O double bonding than WO_6_ bending mode was observed, implying an increase in the number of W^6+^ ions and a structural change to WO_4_ tetrahedra was more dominant than to WO_6_ octahedra. Thus, the CaWO_4_ crystal phase composed of WO_4_ tetrahedra would be precipitated after the heat-treatment, instead of the Ca_3_WO_6_ crystal phase with WO_6_ octahedra.

To understand the structural changes of 33CaO-67B_2_O_3_ glass by Ca_3_WO_6_ addition, more discussion can be given in the following: The borate glass matrix was composed of ring-type metaborates and diborates with bridging oxygens at lower Ca_3_WO_6_ concentration. However, the addition of more Ca_3_WO_6_ contents promoted the structural conversion of them to orthborates (BO_3_)^3−^ with three non-bridging oxygens, as shown in Figure 7a because the strong ionic field of Ca^2+^ ion is enough to break B-O-B networks of metaborate/diborate and produces the isolated structures of orthoborate. Figure 7a also elucidated a larger number of tungsten oxides of covalent WO_4_ (No.7) working as a network formerly in comparison to WO_6_ (No.1) working as a network modifier. This is interpreted so that the ionic nature of Ca^2+^ ions mainly affected the borate networks, while more covalent WO_4_ entities can take part in the construction of a glass network with the borate structures with bridging oxygens to maintain the glassy structures after the quite rapid quenching. Even at *x* = 16 mol%, the B_2_O_3_ concentration was higher than WO_3_ (B_2_O_3_/WO_3_ = 3.51) and the increasing orthoborates would make the vitrification difficult from the viewpoint of structural chemistry. It should be borne in mind that the oxygen basicity was also increased with the introduction of Ca^2+^ ions as CaO from Ca_3_WO_6_ content, which promoted the production of W=O double bonds as well as O-W-O linkages. The W=O bond is not a part of the WO_6_, but the WO_4_ structure. From these considerations, it can therefore be deduced that the formation of WO_4_ structures may support stabilizing their glassy state and also be important to model the structural changes of the calcium borate glasses obtained by the rapid water quenching with the addition of more Ca_3_WO_6_ contents.

### 3.6. Luminescence Properties

Figure 8 displays PL spectra and asymmetry ratio (∧) of 15 mol% Ca_2.98_Eu_0.02_WO_6_ glass, glass-ceramic, and Ca_0.98_Eu_0.02_WO_4_ normalized by PL intensity at 593nm. Three peaks of ~578 nm, ~590 nm, and ~615 nm originate from Eu^3+^ luminescence assignable to ^5^D_0_–^7^F_0_, ^5^D_0_–^7^F_1,_ and ^5^D_0_–^7^F_2_ transitions, respectively [27]. Asymmetry ratio, which is defined as a PL intensity ratio of electric dipole ^5^D_0_–^7^F_2_ and magnetic dipole ^5^D_0_–^7^F_1_ intensities (∧ = *I*(^5^D_0_–^7^F_2_)/*I*(^5^D_0_–^7^F_1_)), estimated for discussion on the degree in asymmetry of local structure around Eu^3+^ ions [28,29,30]. It is well known that ^5^D_0_–^7^F_2_ transition intensity is much affected by local asymmetry around Eu^3+^ and ^5^D_0_–^7^F_1_ transition is independent of the local structure because of the respective natures of electric and magnetic dipole transitions [31]. Asymmetry ratios of 15mol% Ca_2.98_Eu_0.02_WO_6_ glass, glass-ceramic and Ca_0.98_Eu_0.02_WO_4_ are estimated as 3.0, 2.1, and 7.5, respectively. It is found that the asymmetry ratio of 15 mol% Ca_2.98_Eu_0.02_WO_6_ sample is lower in the glassy state than that of the crystal and decreased through the heat-treatment, although despite CaWO_4_ crystals, were precipitated in the glass-ceramics. The result suggests that the heat-treatment rearranged Eu^3+^ environments in the sample and eventually allowed the local symmetry of ligand structures around Eu^3+^ ion to be improved, and most of the Eu^3+^ ions were not positioned in the CaWO_4_ crystal phase precipitated but in the glass matrix. This is interpreted such that Ca and W ions were used for the precipitation of CaWO_4_ in a borate glass matrix and Eu^3+^ ions were forced to be located in a more ionic matrix of calcium borate glass with less W content. This is matched with the results of the Raman investigation. It is assumed that the glass structure becomes a more stable phase, because of ions transfer by heating.

To estimate how the doped Eu^3+^ ions were incorporated in the parent glass or the crystalline phase during the transformation of glass to the glass–ceramic system, Eu^3+ 5^D_0_ decay curves were examined and shown in Figure 9. It is seen that the luminescence decays were almost identical for the pristine glass and the glass-ceramic, while the Eu^3+^ ions in CaWO_4_ crystal exhibited a faster ^5^D_0_ decay curve. The evaluated lifetimes were 6.248 ± 0.007, 5.735 ± 0.008, and 1.479 ± 0.001 ms for the pristine glass, glass-ceramic, and Ca_0.98_Eu_0.02_WO_4_ crystal, respectively. The PL lifetime is found to start to decrease a bit by the formation of the glass-ceramic, however, the values for the glass and glass-ceramic were almost the same, ~6 ms. However, the crystal Ca_0.98_Eu_0.02_WO_4_ had a shorter lifetime of ~1.5 ms. The difference in the lifetime between the glass/glass-ceramic and the crystal corresponds to the behavior of the asymmetry ratios obtained from the PL data. As seen in a higher resolution PL detection (Appendix A in Appendix A), sharp PL lines in ^5^D_0_-^7^F_2_ transition were more or less observed for the glass-ceramic, from which it is imagined that a part of Eu^3+^ ions could enter the CaWO_4_ crystal but the majority of the ions was still in the glass matrix. In general, a higher asymmetry ratio of Eu^3+ 5^D_0_-^7^F_1,2_ luminescence means an increase in the probability of ^5^D_0_-^7^F_2_ transition in electric dipole nature against that of ^5^D_0_-^7^F_1_ transition in magnetic dipole nature, which will induce the enhancement of total transition probability from ^5^D_0_ level and give faster ^5^D_0_ luminescence lifetime. This can well explain the difference in the behavior of the observed spectra and PL decays between the glass/glass-ceramic and the crystal. From the results, it can be mentioned that Eu^3+^ ions were still positioned in the glassy matrix even after the crystallization. More to be mentioned finally, the slight decrease in the PL lifetime and the decrease in the asymmetry ratio for the glass-ceramic in comparison to the pristine glass imply the presence of ion-ion interaction between Eu^3+^ ions and/or non-radiative transition in the glass-ceramic with higher symmetric sites for Eu^3+^ ions.

## 4. Conclusions

Synthesis of glass-ceramic, including the CaWO_4_ phase, was succeeded by the heat-treatment at a temperature higher than *T*_g_ for (100 − *x*) (33CaB) − *x*Ca_3_WO_6_ (*x* = 2–15 mol%). XRD data clarified the precipitation of CaWO_4_ crystals in *x* = 2, 12, and 15 mol% samples by the heat-treatment. The 2mol% Ca_3_WO_6_ glass-ceramic sample included crystal phases of CaB_4_O_7_ and CaB_2_O_4_. DSC curves and Raman spectra were examined to understand the crystallization mechanism of CaWO_4_, CaB_4_O_7,_ and CaB_2_O_4_ in the glasses. The stability of glassy state ΔT increased in the range from *x* = 0 to 2 mol% and then decreased in the range of higher than 4mol% Ca_3_WO_6_ content. The above change corresponded with the numbers of glass network former of borates with bridging oxygens, which was elucidated by Raman spectra. The result suggested that the decreasing borate ring structures helped the crystalization of CaWO_4_ without B_2_O_3_ components by the heat-treatment. Contrary to the structural evolution of borates, the W=O double bondings in WO_4_ tetrahedra were significantly increased with the Ca_3_WO_6_ content, while the WO_6_ octahedra bending vibrations tended to slowly increase, which was also found by Raman spectroscopy. This behavior was a cause of the induction of CaWO_4_ composed of WO_4_ tetrahedral units by the heat-treatment instead of Ca_3_WO_6_ with WO_6_ octahedral units. PL investigation in the scope of asymmetry ratio of Eu^3+^ PL spectra demonstrated that Eu^3+^ ions doped in the 15mol% sample were surrounded by a ligand structure with higher local symmetry after the heat-treatment than Eu^3+^ ions in the pristine glass or CaWO_4_ crystals, indicating that the heat-treatment forced Eu^3+^ ions to be located in the more ionic matrix of calcium borate glass with less W content.

## Figures and Tables

**Figure 1 materials-14-00952-f001:**
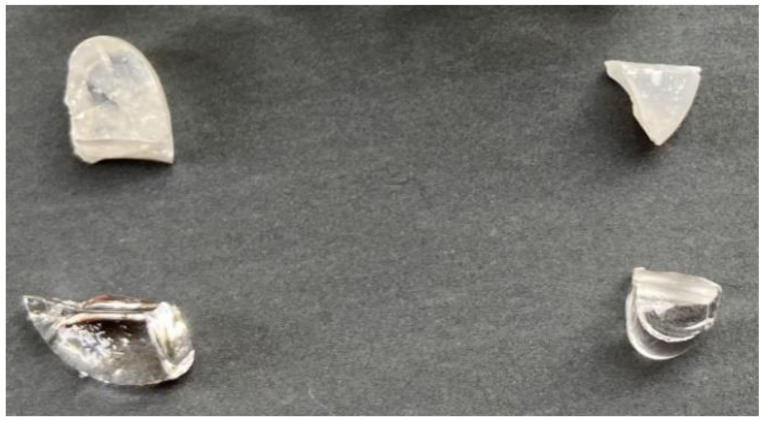
Photo-image of (100 − *x*) (33CaO-67B_2_O_3_) − *x* Ca_3_WO_6_ (*x* = 4 (Left) and 8 (Right)) glass before (Downward) and after (Upward) heat-treatment.

**Figure 2 materials-14-00952-f002:**
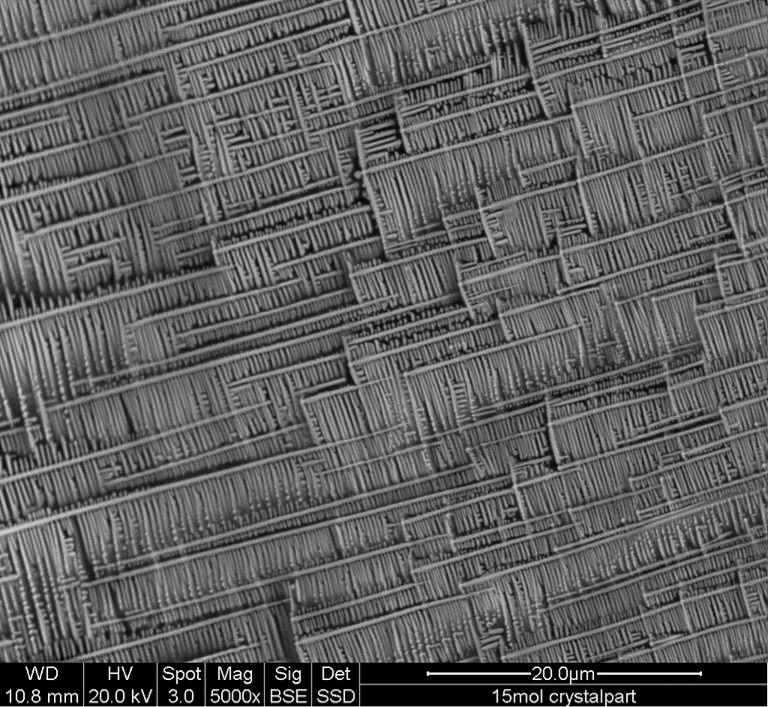
SEM image of the crystal part of 85 (33CaO-67B_2_O_3_) – *15* Ca_3_WO_6_ glass.

**Figure 3 materials-14-00952-f003:**
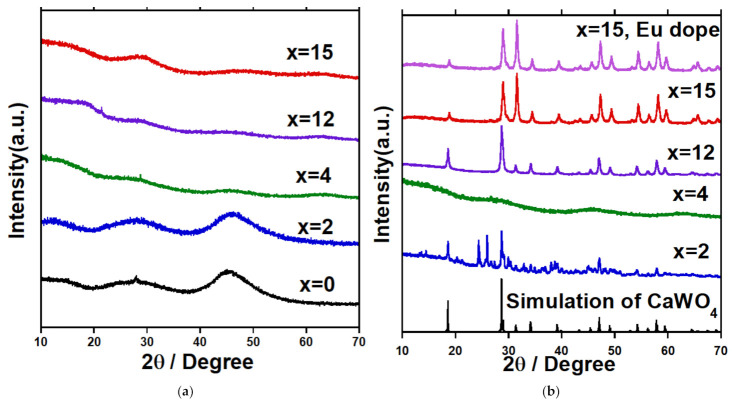
XRD patterns of (**a**) (100 − *x*) (33CaO-67B_2_O_3_) − *x* Ca_3_WO_6_ glass samples and (**b**) (100 − *x*) (33CaO-67B_2_O_3_) − *x* Ca_3_WO_6_ samples after heat-treatment at higher than each *T*_g_ through to 90 h.

**Figure 4 materials-14-00952-f004:**
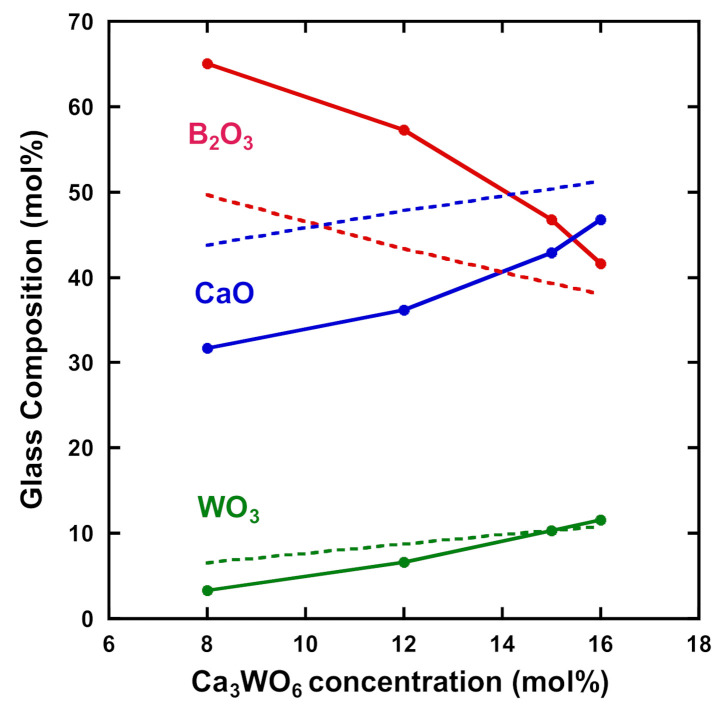
Glass composition estimated by EDS measurements for (100 − *x*) (33CaO-67B_2_O_3_) − *x* Ca_3_WO_6_ (dashed and solid lines show theoretical and experimental contents, respectively).

**Figure 5 materials-14-00952-f005:**
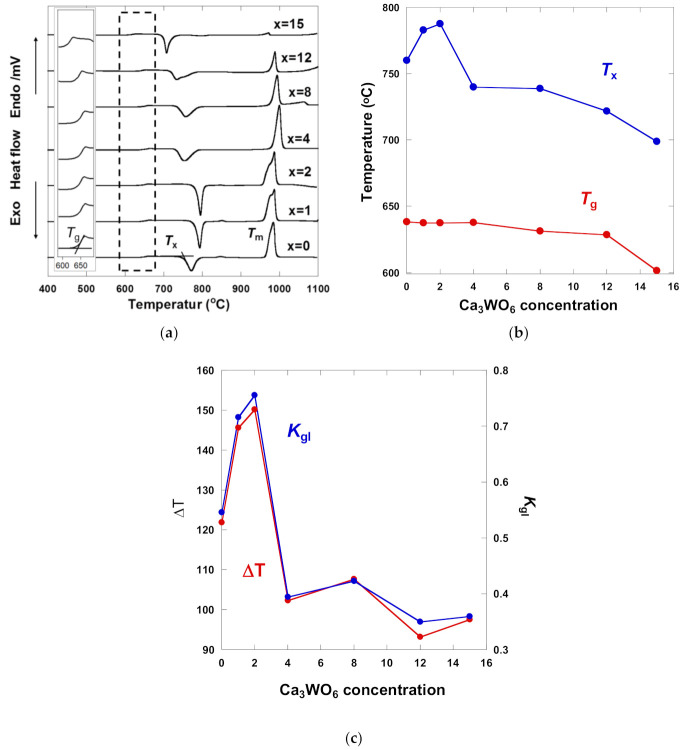
(**a**) DSC curves of (100 − *x*) (33CaO-67B_2_O_3_) − *x* Ca_3_WO_6_ glasses with an insertion of the expanded figure around *T*_g_, (**b**) The values of *T*_g_ and *T*_x_, and (**c**) ΔT and *K*_gl_ as a function of Ca_3_WO_6_ concentration.

**Figure 6 materials-14-00952-f006:**
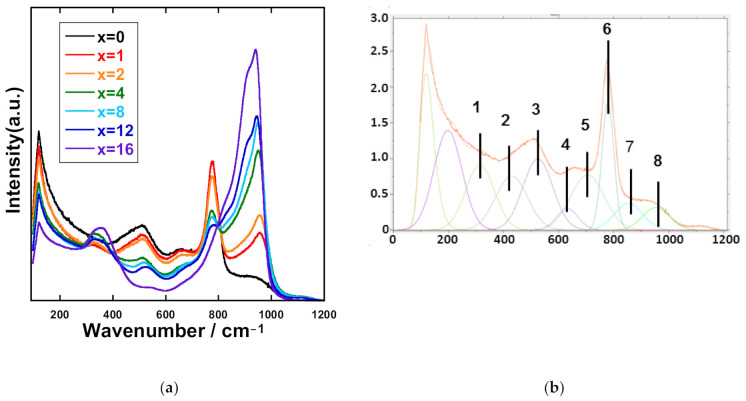
(**a**) Area normalized Raman spectra of 0–16 mol% Ca_3_WO_6_ glass samples and (**b**) deconvoluted Raman spectra of 33CaO–67B_2_O_3_ glass.

**Figure 7 materials-14-00952-f007:**
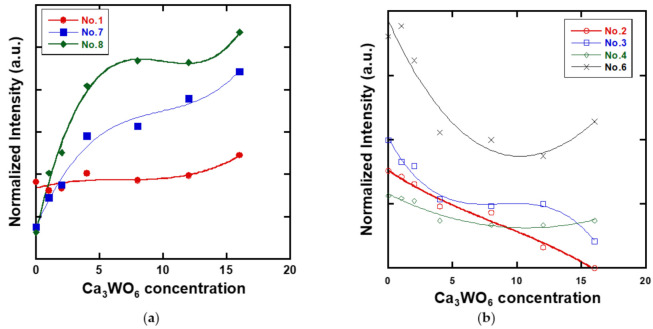
The decomposition peak intensity of (**a**) Nos.1, 7, and 8, and (**b**) Nos.2, 3, 4, and 6 with Ca_3_WO_6_ concentration.

**Figure 8 materials-14-00952-f008:**
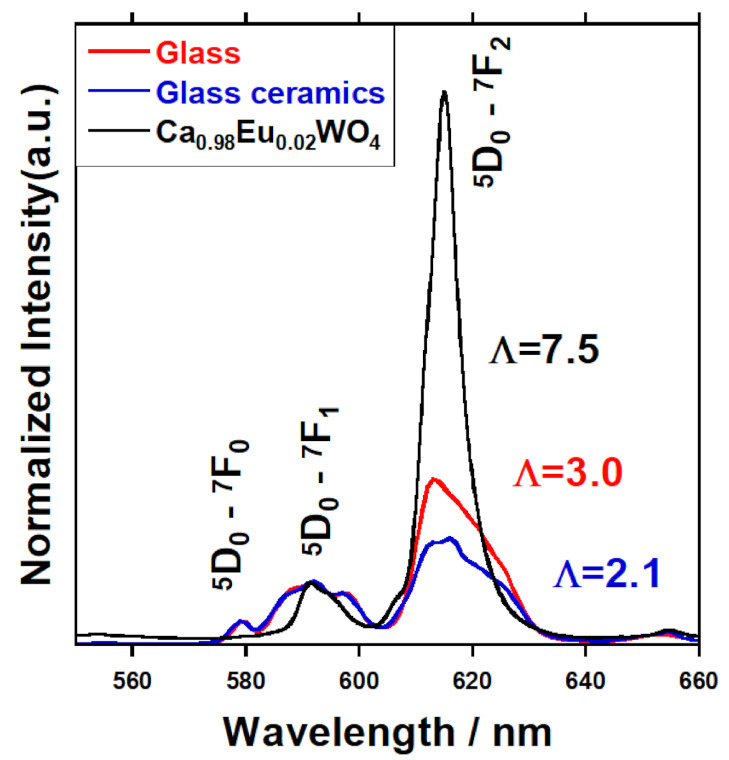
PL spectra and asymmetry ratio (∧) of 15mol% Ca_2.98_Eu_0.02_WO_6_ samples (pristine glass and glass-ceramic) and Ca_0.98_Eu_0.02_WO_4_ crystal, normalized by the 593 nm PL intensity.

**Figure 9 materials-14-00952-f009:**
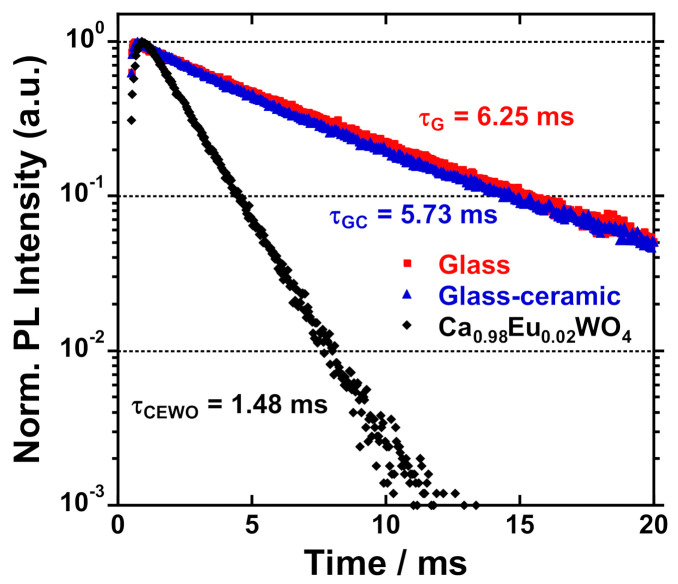
Eu^3+^ PL decay curves of the 15mol% Ca_2.98_Eu_0.02_WO_6_ samples (the pristine glass and glass-ceramic) and Ca_0.98_Eu_0.02_WO_4_ crystal (λ_ex_ = 394nm, λ_em_ = 613nm). The evaluated PL lifetime was τ_G_ = 6.248 ± 0.007 ms (the pristine glass), τ_GC_ = 5.735 ± 0.008 ms (the glass-ceramic), and τ_CEWO_ = 1.479 ± 0.001 ms (the Ca_0.98_Eu_0.02_WO_4_ crystal).

**Table 1 materials-14-00952-t001:** Summary of sample preparation: Quenching condition (N.Q.: Normal Quenching, W.Q.: Water Quenching), Ratio of Glass (G) and Crystal (C) inside the species quenched, Location of crystals (SurF.: Surface, IntF.:Interface between glass and crucible, V.C.: Volume Crystallization, n.d.: not detected), Results of XRD and Glass Transition temperature Tg for the obtained glasses (pristine), Heating condition (Temperature and Time) for fabricating glass-ceramics (GC), Results of XRD and Raman experiments, and Detected phases of crystals after the heat-treatment, for (100 − *x*)(33CaO-67B_2_O_3_) − *x*Ca_3_WO_6_(or Ca_2.98_Eu_0.02_WO_6_) samples. (-: no data; n.a.: not applied).

x	Quenching Condition	Ratio (G:C)	Location of Crystals	XRD of the Pristine	Tg	Heating Condition (Temp./Time)	XRD after Heating	Raman after Heating (Appendix A)	Phases of Crystals
0	N.Q.	10:0	n.d.	G	638.3 °C	800 °C /40 min	-	GC	CaB_4_O_7_, CaB_2_O_4_
1	N.Q.	10:0	n.d.	G	637.5 °C	800 °C /40 min	-	GC	CaB_4_O_7_, CaB_2_O_4_, CaWO_4_
2	N.Q.	10:0	n.d.	G	637.5 °C	660 °C /90 h	GC	GC	CaB_4_O_7_, CaB_2_O_4_, CaWO_4_
4	N.Q.	10:0	n.d.	G	637.7 °C	665 °C /90 h	G	GC+G	CaWO_4_
8	N.Q.	~10:0	SurF.	G	631.3 °C	665 °C /90 h	-	GC	CaWO_4_
12	W.Q.	9:1	IntF.	G	628.7 °C	660 °C /90 h	GC	GC	CaWO_4_
15	W.Q.	9:1	IntF.	G	601.4 °C	635 °C /90 h	GC	GC	CaWO_4_
16	W.Q.	6:4	V.C.	GC	-	n.a.	-	-	-
15, Eu^3+^	W.Q.	9:1	IntF.	G	-	635 °C /90 h	GC	-	CaWO_4_

## Data Availability

Data is contained within the article.

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
