# Peer review of "Thermal Evolutions to Glass-Ceramics Bearing Calcium Tungstate Crystals in Borate Glasses Doped with Photoluminescent Eu3+ Ions"

_materials, 2021, doi:10.3390/ma14040952_

Round 1

Reviewer 1 Report

First, the English language is such that in some places it is not clear what the authors mean. It starts just after the list of affiliations. There we read "Both authors contributed equally to this manuscript", although there are 4 authors.

In order not to be unfounded, the next example (lines 106-108): “The 12 and 15mol% samples were crystallized at the interface between the glass and the crucible, as confirmed after the water quenching. However, the glass part of the samples in the vicinity of the crystals remained clear and transparent.”  What does "confirmed after the water quenching" mean? What? if "water quenching" is not used, there will be no crystallization at the interface? Is "water quenching" exactly what made it possible to reveal (confirm) this crystallization? And what does “glass part of the samples in the vicinity of the crystals” mean?  

Concerning scientific information. The authors generally do not indicate definite temperatures used in heat treatments of glasses. The authors confine themselves to the phrase "the post heat-treatment at a temperature higher than Tg". Well, even if the exact temperature of the heat treatment is not indicated, then it is necessary to indicate, at least, the excess of this temperature relative to Tg. And then, what does "post heat treatment" mean? Maybe just "heat treatment" is enough, because no "before heat treatment" is mentioned in the text.

The entire text is embedded with stylistic blunders and inadequate terminology. I see no point in reading the article in detail until the language and style have been revised.

Author Response

First of all, thank you for reviewing our paper. The comments and questions given by the reviewers are all fruitful for us. We read them carefully and revised our manuscript. Here we would like to reply to each of them. Revisions were marked in red ink in the revised manuscript. 

Reviewer’s comment #1-1

First, the English language is such that in some places it is not clear what the authors mean. It starts just after the list of affiliations. There we read "Both authors contributed equally to this manuscript", although there are 4 authors.

Reply to #1-1

Thank you for reviewing our manuscript. This sentence was omitted. In the revised manuscript, the total number of authors becomes five because the added member contributes to this work by the acquisition of PL decay curves, which are inserted as Fig.9 in the revised manuscript.

Reviewer’s comment #1-2

In order not to be unfounded, the next example (lines 106-108): “The 12 and 15mol% samples were crystallized at the interface between the glass and the crucible, as confirmed after the water quenching. However, the glass part of the samples in the vicinity of the crystals remained clear and transparent.” What does "confirmed after the water quenching" mean? What? if "water quenching" is not used, there will be no crystallization at the interface? Is "water quenching" exactly what made it possible to reveal (confirm) this crystallization? And what does “glass part of the samples in the vicinity of the crystals” mean?

Reply to #1-2

Thank you for your comments. There were confusing sentences that should be revised. For x=0~8 mol%, a normal quenching speed was enough to obtain the transparent glass samples, while quit rapid quenching into water (called “water quenching” here) was needed over x=12 mol%. However, the obtained glasses with x=12~16 mol% still had a part of crystals even after water quenching. It was because the quenching was conducted so that the crucible with glass melt was put into water from the bottom of the crucible and roughly 80% of the crucible was sunk in the water bath. The top of the melt was not covered with water to avoid unexpected influences of water molecules on the structure of the obtained glasses. Before the crucible was completely cooled down to room temperature, the crucible was taken from the water bath, resulting in that the super-cooled liquid might be kept warm to allow it to be crystallized in part. The crystallized portion was seen only between the interface of glass melt and crucible for x = 12 and 15 mol% although it was small. The volume ratio of glass and crystals was roughly 9:1 for x = 12 and 15 mol%, and 6:4 for x=16 mol%. The crystal for x=15 mol% was examined for SEM observation, which was shown in Fig.3 in the first manuscript (or Fig.2 in the revised manuscript). The crystallized parts for 12 and 15 mol% were carefully removed, and the remaining glass species, which were clear and transparent, were used for the heat treatment.

The sentences indicated by the reviewer were omitted and alternatively the following sentences and (new) Table 1 were added in the revised manuscript. (See also “Reply to #1-3”)

“Although a normal quenching rate was applied for x=0~8 mol% to obtain the transparent glass samples, quit rapid quenching into water (called “water quenching” here) was needed over x=12 mol%. However, even after the water quenching, the obtained glasses at x=12~16 mol% still had a part of crystals. It was because the quenching was conducted so that the crucible with glass melt was put into water from the bottom of the crucible and roughly 80% of the crucible was sunk in the water bath. The top of the melt was not covered with water to avoid unexpected influences of water molecules on the structure of the obtained borate glasses. If the crucible was not completely cooled down to room temperature and then the crucible was taken from the water bath, the supercooled liquid was resultantly kept warm to allow it to be partially crystallized. Nevertheless, the crystallized portion was only seen between the interface of glass melt and crucible for x = 12 and 15 mol% and was found to be small. The volume ratio of glass and crystals was about 9:1 for x = 12 and 15 mol%, and about 6:4 for x=16 mol%, as shown in Table 1.” “Toward further experiments, the crystallized parts for 12 and 15 mol% were carefully removed and then clear and transparent glassy parts were extracted. On the other hand, at x = 16 mol%, serious crystallization occurred and thus it was not used for the heat treatment.”

Reviewer’s comment #1-3

Concerning scientific information. The authors generally do not indicate definite temperatures used in heat treatments of glasses. The authors confine themselves to the phrase "the post heat-treatment at a temperature higher than Tg ".Well, even if the exact temperature of the heat treatment is not indicated, then it is necessary to indicate, at least, the excess of this temperature relative to Tg.

Reply to #1-3

To easily see the temperatures used for the heat treatment in comparison with Tg, the heat treatment conditions and Tg are tabulated as Table 1 in the revised manuscript. The heat treatment temperatures used for obtaining glass-ceramics were also given in “2.1 Glass Synthesis & Crystallization” in the maintext.

Table 1. Summary of sample preparation: Quenching condition (N.Q.: Normal Quenching, W.Q.: Water Quenching), Ratio of Glass (G) and Crystal (C) inside the species quenched, Location of crystals (SurF.: Surface, IntF.:Interface between glass and crucible, V.C.: Volume Crystallization, n.d.: not detected), Results of XRD and Glass Transition temperature Tg for the obtained glasses (pristine), Heating condition (Temperature and Time) for fabricating glass-ceramics (GC), Results of XRD and Raman experiments, and Detected phases of crystals after the heat treatment, for (100-x)(33CaO-67B2O3)-xCa3WO6(or Ca2.98Eu0.02WO6) samples. (-: no data; n.a.: not applied)

x

Quenching condition

Ratio

(G:C)

Location of crystals

XRD of the pristine

Tg

Heating condition (Temp./Time)

XRD after heating (Fig.3(b))

Raman after heating (Fig.S2)

Phases of Crystals

0

N.Q.

10:0

n.d.

G

638.3 oC

800oC

/ 40min

-

GC

CaB4O7, CaB2O4

1

N.Q.

10:0

n.d.

G

637.5 oC

800oC

/ 40min

-

GC

CaB4O7, CaB2O4, CaWO4

2

N.Q.

10:0

n.d.

G

637.5 oC

660oC

/ 90h

GC

GC

CaB4O7, CaB2O4, CaWO4

4

N.Q.

10:0

n.d.

G

637.7 oC

665 oC

/ 90h

G

GC+G

CaWO4

8

N.Q.

~10:0

SurF.

G

631.3 oC

665 oC

/ 90h

-

GC

CaWO4

12

W.Q.

9:1

IntF.

G

628.7 oC

660 oC

/ 90h

GC

GC

CaWO4

15

W.Q.

9:1

IntF.

G

601.4 oC

635 oC

/ 90h

GC

GC

CaWO4

16

W.Q.

6:4

V.C.

GC

-

n.a.

-

-

-

15, Eu3+

W.Q.

9:1

IntF.

G

-

635 oC

/ 90h

GC

-

CaWO4

Reviewer’s comment #1-4
And then, what does "post heat treatment" mean? Maybe just "heat treatment" is enough, because no "before heat treatment" is mentioned in the text.

Reply to #1-4
We avoided using the phrase “post heat treatment” and simply used “heat-treatment”.

Reviewer’s comment #1-5
The entire text is embedded with stylistic blunders and inadequate terminology. I see no point in reading the article in detail until the language and style have been revised.

Reply to #1-5
The manuscript was revised greatly along with the reviewer’s suggestions. Thank you again.

Reviewer 2 Report

  1. “No. 2 and 3” change to “Nos. 2 and 3”. Same for others.
  2. Note that the figures appear in the order of the articles. For example, Figs 3 and 4.
  3. The authors declared that “bridging oxygen and non-bridging oxygen decreased and increased with Ca3WO6 concentration, respectively, shows the host glass structures were strongly affected by Ca2+ ions working as a network modifier with the Ca3WO6 additions.”. Why?
  4. The peak intensity of No. 8 is larger than Nos. 1 and 7 exhibited an increasing tendency with Ca3WO6 concentration. Why?

Author Response

First of all, thank you for reviewing our paper. The comments and questions given by the reviewers are all fruitful for us. We read them carefully and revised our manuscript. Here we would like to reply to each of them. Revisions were marked in red ink in the revised manuscript. 

Reviewer’s comment #2-1

  1. “No. 2 and 3” change to “Nos. 2 and 3”. Same for others.

Reply to #2-1

We revised the manuscript along with the suggestions.

Reviewer’s comment #2-2

  1. Note that the figures appear in the order of the articles. For example, Figs 3 and 4.

Reply to #2-2

In the revised manuscript, the organization of the manuscript, as well as figures, was re-considered so that Fig.3(a) for SEM image was moved to the position of Fig.2, while Fig.3(b) for EDS analysis was renumbered as Fig.4. Correspondingly the original Figs.2, 4 to 7 was also renumbered as Figs.3, 5 to 8. By this reorganization, the impacts of the water quenching, including the partial crystallization for the higher Ca3WO6 samples and the SEM image, can be discussed in the same subsection of 3.1. And in the revised manuscript, the data for the glasses were seen in the order of XRD (Fig.3(a)), EDS analysis (Fig.4), Thermal properties (Fig.5), and Raman (Figs.6 and 7). And the data for the glass-ceramics were shown in the order of XRD (Fig.3(b)), Raman (Fig.S2), PL (Fig.8), and (new) PL decay (Fig.9)

As for the obtained SEM image, the following sentences were added in 3.1 in the revised manuscript.

.. If the crucible was not completely cooled down to room temperature and then the crucible was taken from the water bath, the supercooled liquid was resultantly kept warm to allow it to be crystallized. Nevertheless, the crystallized portion was just only seen between the interface of glass melt and crucible for x = 12 and 15 mol% and was found to be small. The volume ratio of glass and crystals was about 9:1 for x = 12 and 15 mol%, and about 6:4 for x=16 mol%, as shown in Table 1. (Other possibilities will be discussed in 3-3. EDS elementary analysis for 33CaB-Ca3WO6 glasses.) Figure 2 shows an SEM image of the crystal part of the 15 mol% sample of 85(33CaO-67B2O3)–15Ca3WO6. The observed morphology of the crystal is so unique, and the SEM image clarifies the line structure of the surface of the crystal, indicating uniaxial nuclear growth of the tetragonal CaWO4 phase.

And in 3-3. EDS elementary analysis for 33CaB-Ca3WO6 glasses, the sentences were revised as follows.

“Figure 4 shows the compositions (B2O3, CaO, and WO3) of the 33CaB-Ca3WO6 glasses (glass parts) for x = 8 to 16 mol%, estimated by EDS measurements. Qualitatively the decrease in B2O3 content and the increase in CaO and WO3 with the addition of Ca3WO6 are matched with the variations of the theoretical contents. However, the experimental values of CaO and WO3 contents became lower than the theoretical ones for the respective components. On the other hand, the experimental value of B2O3 content was higher than the theoretical one. It may be caused by possible evaporation of CaO and WO3 components during melting because the addition of Ca3WO6 to 33CaB glass required the higher melting temperature of 1400oC.”

Reviewer’s comment #2-3

  1. The authors declared that “bridging oxygen and nonbridging oxygen decreased and increased with

Ca3WO6 concentration, respectively, shows the host glass structures were strongly affected by Ca2+ ions working as a network modifier with the Ca3WO6 additions.”. Why?

Reply to #2-3

The borate glass matrix was composed of ring-type metaborates and diborates with bridging oxygens at lower Ca3WO6 concentration. However, the addition of more Ca3WO6 contents promoted the structural conversion of them to orthborates with three non-bridging oxygens, as shown in Fig.7(a) because the strong ionic field of Ca2+ ion is enough to break B-O-B networks of metaborate/diborate and produce the isolated structures of orthoborate. Fig.7(a) also elucidated a larger number of tungsten oxides of covalent WO4 (No.7) working as a network former in comparison with WO6 (No.1) working as a network modifier. This is interpreted so that the ionic nature of Ca2+ ions affected mainly the borate networks while more covalent WO4 entities can take part in the construction of glass network with the borate structures with bridging oxygens to maintain the glassy structures after the quite rapid quenching. Even at x=16 mol%, the B2O3 concentration was higher than WO3 (B2O3/WO3 = 3.51) and the increasing orthoborates would make the vitrification difficult from the viewpoint of structural chemistry. As mentioned above, the formation of WO4 structures may support stabilizing their glassy state. These are important to understand the above structural changes of the calcium borate glasses by the addition of Ca3WO6 content.

Reviewer’s comment #2-4

  1. The peak intensity of No. 8 is larger than Nos. 1 and 7 exhibited an increasing tendency with Ca3WO6 concentration. Why?

Reply to #2-4

The peak No. 8 was assigned to B-O vibration in orthoborate structure of three-fold coordinated boron with three non-bridging oxygen while the Nos.1 and 7 mainly came from O-W-O bending modes of WO6 clusters and a W=O stretching motion of WO4 unit, respectively, as mentioned in the maintext. The increase in the No.8 peak intensity means the presence of more isolated borate structures (BO3)3- induced with the addition of Ca2+ ions due to the reason mentioned in Reply to the comment #2-3. The oxygen basicity was also increased with the introduction of Ca2+ ions as CaO from Ca3WO6 content, which promoted to produce W=O double bonds as well as O-W-O linkages. The W=O bond is not a part of WO6 but WO4 structure. That is a key to understand the possible precipitation of CaWO4 crystal in glass-ceramic with x value over 4 mole%.

The above discussion in Replies to #2-3 and #2-4 was arranged and added at the end of the subsection of “3.5 Raman spectroscopy” in the revised manuscript as follows:

“To understand the structural changes of 33CaO-67B2O3 glass by Ca3WO6 addition, more discussion can be given in the following: The borate glass matrix was composed of ring-type metaborates and diborates with bridging oxygens at lower Ca3WO6 concentration. However, the addition of more Ca3WO6 contents promoted the structural conversion of them to orthborates (BO3)3- with three non-bridging oxygens, as shown in Fig.7(a) because the strong ionic field of Ca2+ ion is enough to break B-O-B networks of metaborate/diborate and produce the isolated structures of orthoborate. Fig.7(a) also elucidated a larger number of tungsten oxides of covalent WO4 (No.7) working as a network former in comparison with WO6 (No.1) working as a network modifier. This is interpreted so that the ionic nature of Ca2+ ions affected mainly the borate networks while more covalent WO4 entities can take part in the construction of glass network with the borate structures with bridging oxygens to maintain the glassy structures after the quite rapid quenching. Even at x=16 mol%, the B2O3 concentration was higher than WO3 (B2O3/WO3 = 3.51) and the increasing orthoborates would make the vitrification difficult from the viewpoint of structural chemistry. It should be borne in mind that the oxygen basicity was also increased with the introduction of Ca2+ ions as CaO from Ca3WO6 content, which promoted to produce W=O double bonds as well as O-W-O linkages. The W=O bond is not a part of WO6 but WO4 structure. From these considerations, it can be deduced that the formation of WO4 structures may support stabilizing their glassy state and be also important to model the structural changes of the calcium borate glasses obtained by the rapid water quenching with the addition of more Ca3WO6 contents.”

Reviewer 3 Report

Luminescent glass-ceramics containing CaWO4 crystals and Eu3+ ions have been examined in details. In general, the manuscript is interesting and should be published in the Materials after minor revision. Based on emission spectra measurements and the asymmetry ratios of red to orange bands, the Authors suggest that "... most of the Eu3+ ions were not positioned in the CaWO4 crystal phase precipitated but in the glass matrix" (Part 3.5). In fact, red-to-orange fluorescence intensity ratio gives information about the nearest surrounding (higher or lower local symmetry) of europium ions and bonding (more ionic or covalent) between europium and surrounding ligands. The changes in luminescence lifetimes 5D0 (Eu3+) should be examined in order to estimate transformation of glass to glass-ceramic system and the incorporation of europium ions in parent glass and/or crystalline phase. Luminescence decay analysis is strongly recommended.

Author Response

First of all, thank you for reviewing our paper. The comments and questions given by the reviewers are all fruitful for us. We read them carefully and revised our manuscript. Here we would like to reply to each of them. Revisions were marked in red ink in the revised manuscript. 

Reviewer’s comment #3-1

Luminescent glass-ceramics containing CaWO4 crystals and Eu3+ ions have been examined in details. In general, the manuscript is interesting and should be published in the Materials after minor revision.

Reply to #3-1

Thank you for reviewing our paper and giving us fruitful comments. The manuscript was revised by the addition of new data on Eu3+ PL lifetimes, and the corresponding explanation was given in the revised manuscript (See also Reply to #3-2).

Reviewer’s comment #3-2

Based on emission spectra measurements and the asymmetry ratios of red to orange bands, the Authors suggest that "... most of the Eu3+ ions were not positioned in the CaWO4 crystal phase precipitated but in the glass matrix" (Part 3.5). In fact, red-to-orange fluorescence intensity ratio gives information about the nearest surrounding (higher or lower local symmetry) of europium ions and bonding (more ionic or covalent) between europium and surrounding ligands. The changes in luminescence lifetimes 5D0 (Eu3+) should be examined in order to estimate transformation of glass to glass-ceramic system and the incorporation of europium ions in parent glass and/or crystalline phase. Luminescence decay analysis is strongly recommended.

Reply to #3-2

Along with the reviewer’s suggestions, we measured luminescence lifetime 5D0 of the glass and glass-ceramic as well as that of the Ca0.98Eu0.02WO6 crystal. The data are added in the revised manuscript as Fig.9, which elucidates that the Eu3+ 5D0 lifetimes of the glass and glass-ceramic are identical and that the lifetime of Eu3+ PL in CaWO4 crystal was shorter ~1.5 ms than those of the glass and glass-ceramic ~ 6 ms. This observation is well correspondent to the results on the asymmetry ratios obtained from the PL data and validates our proposition that Eu3+ ions are still positioned even after the crystallization.

The following sentences and Eu3+ PL decay data (Fig.9) are added in the revised manuscript.

“To estimate how the doped Eu3+ ions were incorporated in the parent glass or the crystalline phase during the transformation of glass to glass-ceramic system, Eu3+ 5D0 decay curves were examined and shown in Fig.9. It is seen that the luminescence decays were almost identical for the pristine glass and the glass-ceramic, while the Eu3+ ions in CaWO4 crystal exhibited a faster 5D0 decay curve. The evaluated lifetimes were 6.2480.007, 5.7350.008, and 1.4790.001 ms for the pristine glass, glass-ceramic, and Ca0.98Eu0.02WO4 crystal, respectively. The PL lifetime is found to start to decrease a bit by the formation of the glass-ceramic and however, the values for the glass and glass-ceramic were almost the same, ~ 6 ms. On the other hand, the crystal Ca0.98Eu0.02WO4 had a shorter lifetime of ~1.5 ms. The difference in the lifetime between the glass/glass-ceramic and the crystal is well correspondent to the behavior of the asymmetry ratios obtained from the PL data. As seen in a higher resolution PL detection (Figs.S3-4 in Supplementary), sharp PL lines in 5D0-7F2 transition were more or less observed for the glass-ceramic, from which it is imagined that a part of Eu3+ ions could enter to the CaWO4 crystal but the majority of the ions was still in the glass matrix. In general, a higher asymmetry ratio of Eu3+ 5D0-7F1,2 luminescence means an increase in the probability of 5D0-7F2 transition in electric dipole nature against that of 5D0-7F1 transition in magnetic dipole nature, which will induce the enhancement of total transition probability from 5D0 level and give faster 5D0 luminescence lifetime. This can explain the difference in the behavior of the observed spectra and PL decays between the glass/glass-ceramic and the crystal. From the results, it can be mentioned that Eu3+ ions were still positioned in the glassy matrix even after the crystallization. More to be mentioned finally, the slight decrease in the PL lifetime and the decrease in the asymmetry ratio for the glass-ceramic in comparison with the pristine glass imply the presence of ion-ion interaction between Eu3+ ions and/or non-radiative transition in the glass-ceramic with higher symmetric sites for Eu3+ ions. ”

Round 2

Reviewer 2 Report

NO